# Role of Olive Bioactive Compounds in Respiratory Diseases

**DOI:** 10.3390/antiox12061140

**Published:** 2023-05-23

**Authors:** Ubashini Vijakumaran, Neng-Yao Goh, Rabiatul Adawiyah Razali, Nur Atiqah Haizum Abdullah, Muhammad Dain Yazid, Nadiah Sulaiman

**Affiliations:** Centre for Tissue Engineering & Regenerative Medicine, Faculty of Medicine, Universiti Kebangsaan Malaysia, Jalan Yaacob Latif, Cheras, Kuala Lumpur 56000, Malaysia; p110051@siswa.ukm.edu.my (U.V.); gohnengyao@um.edu.my (N.-Y.G.);

**Keywords:** olive, respiratory disease, anti-inflammation, antioxidant, oxidative stress, cancer

## Abstract

Respiratory diseases recently became the leading cause of death worldwide, due to the emergence of COVID-19. The pathogenesis of respiratory diseases is centred around inflammation and oxidative stress. Plant-based alongside synthetic drugs were considered as therapeutics due to their proven nutraceutical value. One such example is the olive, which is a traditional symbol of the MedDiet. Olive bioactive compounds are enriched with antioxidant, anti-inflammatory, anticancer and antiviral properties. However, there are few studies relating to the beneficial effect of olive bioactive compounds on respiratory diseases. A vague understanding of its molecular action, dosage and bioavailability limits its usefulness for clinical trials about respiratory infections. Hence, our review aims to explore olive bioactive compound’s antioxidant, anti-inflammatory and antiviral properties in respiratory disease defence and treatment. Molecular insight into olive compounds’ potential for respiratory system protection against inflammation and ensuing infection is also presented. Olive bioactive compounds mainly protect the respiratory system by subsiding proinflammatory cytokines and oxidative stress.

## 1. Introduction

The growing prevalence of chronic respiratory diseases (CRDs) has increased morbidity and mortality rates worldwide [1]. Chronic respiratory diseases include chronic obstructive pulmonary disease (COPD), asthma, pneumoconiosis, pneumonia, lung cancer, chronic bronchitis, pulmonary sarcoidosis and tuberculosis [2]. COPD causes 81.7% of CRD deaths and is the third-leading cause of death worldwide, killing almost 3.2 million people annually. Meanwhile, pneumonia is the leading cause of death among geriatric (>65 years old, elderly) and paediatric (<5 years old, children) patients [3]. The World Health Organization (WHO) reported that around 6.8 million people’s lives abruptly ended mainly due to respiratory illnesses during the COVID-19 pandemic era [4]. Thus, the management of CRDs was given priority, encompassing the invention of new drugs, vaccines, antibiotics, cortisone, ventilation tools, inhalation therapies and advanced lung surgical intervention [5]. However, developing drug-resistant organism strains and variants make available treatments less effective [6,7]. Hence, more efficient and atoxic drugs are preferable to ease CRD management, especially during the pandemic. Scientific interest is supported by the fact that more than thirty per cent of FDA-approved drugs are of natural origin [8]. Historically, natural-based therapies have long been incorporated into CRD treatment. More than 2000 years ago, drug delivery for respiratory diseases was performed via inhalation therapies in ayurvedic medicine [9]. Scientifically, Oriola et al. reviewed the potential of plant-derived natural chemicals, thus supporting their benefits for common respiratory disease treatment [10]. 

The olive has been one of the most researched plant varieties throughout the decades for its enormous health benefits [11,12]. It is a traditional symbol of Mediterranean culture. This is reflected by a quote from a famous French writer, Georges Duhamel, “There, where the olive tree gives up, is where the Mediterranean ends. The tree of light is the nature and culture of the Mediterranean” [13,14]. The olive fruit and olive oil are the largest products that are commercialised from the olive tree [14], which serve as primary sources of fat in the MedDiet [15]. In 2013, the United Nations Educational, Scientific and Cultural Organization (UNESCO) added the MedDiet to the “Representative List of the Intangible Cultural Heritage of Humanity”. The MedDiet was also specified as being a healthy diet in the 2015–2020 Dietary Guidelines for Americans [16]. Its nutritional values have been correlated with anti-inflammatory [17,18], cardio-protective [19,20,21], anticancer [22,23], anti-ageing [24,25] and neuroprotection [26,27,28] effects. Interestingly, a meta-analysis of cross-sectional studies demonstrated that the MedDiet was associated with longer telomere length and positive ageing [29]. Even though research on olive bioactive compounds has been carried out over the decades, it is much less studied with regard to respiratory diseases. A simple keyword search containing “Olive AND respiratory“ within the past 20 years shows three times fewer articles published compared to “Olive AND cardiovascular”, thus substantiating the hypothesis that the safety, efficacy, and molecular mechanism of olive compounds on respiratory diseases have not been fully comprehended. Therefore, we aimed to review emerging evidence from in vitro, in vivo and clinical studies of olive phytoconstituents in the prevention or impediment of respiratory disease progression. 

## 2. Olive Bioactive Compounds

The olive is botanically known as *Olea europaea* (L.), predominantly found in the Mediterranean Basin [30] and other temperate regions in Asia, Africa and Europe. There are more than 40 species in this genus and, among these species, *Olea europaea* (L.) is the only species being harvested for oil [31]. Phenolic compounds in olive trees are categorised into five groups, as shown in Figure 1 [32]. Oleuropein is the most prominent polyphenol in olive leaves, followed by hydroxytyrosol, luteolin-7-glucosides, apigenin-7-glucosides and verbascoside [33].

Olive oil was first produced in Greece around 1500 years BC in Bronze Age Minoan Crete [34]. Apparently, it was derived from olive fruits by grinding or pressing them either mechanically or chemically. According to the International Olive Oil Council, virgin olive oil (VOO) should be obtained in a mechanical way under thermal conditions (cold pressing), which does not cause any alteration in the oil. Besides this, olives are also not allowed to go through any processing other than filtration, centrifugation, washing and decantation [35]. Extra virgin olive oil (EVOO) has free acidity (0.8 g oleic acid per 100 g), followed by VOO, with about 2 g per 100 g, and ordinary virgin olive oil (OVOO) has 3.3.g per 100 g [36]. The refining method produces refined olive oil from VOO without altering its glyceridic chemical structure. However, it tends to lose its phenolic compounds due to the refining process [37]. Olive oil is blended with refined olive oil and EVOO to make it suitable for consumption [36]. Olive oil is mainly composed of monounsaturated fatty acids (MUFAs) at 55–83%, polyunsaturated fatty acids (PUFAs) at 4–20% and saturated fatty acids (SFAs) at 8–14%. Phenolic molecules such as oleuropein, tyrosol (TY), hydroxytyrosol (HT), ligstroside and oleocanthal make up ~90% of olive total phenols. The saponifiable fraction of olive oil comprises 90.0% to 99.0% of total weight, which is mainly composed of phospholipids, mono-/di- and triacylglycerols [38]. Meanwhile, unsaponifiable fractions are composed of heterogenous compounds which are non-chemically related to fatty acids such as aliphatic alcohol, pigment sterols and so on [39]. Active biological compounds under saponifiable and non-saponifiable fractions present in olive oil are shown in Table 1.

Among the bioactive compounds in olive oil, polyphenols are the key compound that exert olive-derived health benefits. A growing interest in olive oil polyphenols is clearly shown in a review by Finicelli et al., who collectively report the latest clinical trials utilising olive oil polyphenols based on biodistribution, absorption and metabolism [40]. Interestingly, polyphenols from extra virgin olive oil absorb better when consumed with high-fat and fibre-rich foods [14]. The quantity of polyphenols is the first factor that determines the degree of health benefits followed by effective distribution and absorption. Polyphenols largely vary in ordinary olive oil (OO), virgin olive oil (VOO) and extra virgin olive oil (EVOO). EVOO has the lowest acidity in terms of oleic acid, with 0.8 g of acid per 100 g of fat, while OO has no more than 3.3% in total [41]. Hydrophilic phenols in VOO exist in the form of phenolic acids (e.g., hydroxyphenyl acetic acid), phenolic alcohols (e.g., HT and TY), hydroxy-isochrons, flavonoids (e.g., luteolin), secoiridoids (e.g., oleuropein, oleuropein aglycone) and lignans [42]. The quantity of polyphenols depends on the extensiveness of olive processing. Hence, EVOO has the highest polyphenol content compared to refined olive oils, which is about 500 mg/L [43,44]. A remarkably high concentration of polyphenols and flavonoids was also reported in olive by-products, i.e., paste, pomace and aqueous extract [42,45,46]. Therefore, due to minimal processing, EVOO had higher levels of polyphenols and was more readily absorbed than refined olive oil [47]. Table 2 summarises the main phenolic compounds in virgin olive oil. 

Oleuropein is a secoiridoid phenolic compound derived from raw olive leaves and fruits. It is composed of three structural subunits, secoiridoid (elenolic acid), polyphenol (hydroxytyrosol) and a glucose molecule [48]. It is a major olive bioactive compound alongside demethyloleuropein, nuzhenide oleoside, ligstroside and nuzhenide [49], followed by minor tyrosol and hydroxytyrosol. In 1960, Oleuropein was first isolated and characterised by Panizzi, whereas the chiral centres of the secoiridoid were subsequently discovered by Yoshida et al., in 1970 [50]. The content of oleuropein varies as the olive fruit matures, which accumulates in the olive during the first growth phase and is subsequently reduced in the black maturation phase [51]. Oleuropein makes up 14% dry weight of olives, 61.56 g/kg in leaves and 2.8 mg/kg in oil [52,53]. Interestingly, lyophilised olive fruits could retain oleuropein 20 times more (80.3 g/kg) than fresh olive fruits [54], making them suitable as a nutritional supplement. Hydroxytyrosol (HT) and tyrosol are products of oleuropein hydrolytic breakdown. The exorbitant antioxidant effect of HT is derived from its *o*-dihydroxyphenyl moiety. It donates a hydrogen atom to peroxyl-radicals (ROO*) and replaces it with an HT radical (*) [55]. Hydroxytyrosol is the polyphenol that occurs in a large quantity in table olives regardless of the processing method, where about 250–760 mg/kg is present in *the kalamata* olive [56]. 

Fascinatingly, polyphenol content in food could be extracted from Phenol-Explorer which is a comprehensive database that extracts information from published scientific research [57]. Pure HT is the highest in olives, with 74.3 mg/kg in olive oil and 4133 mg/kg in olive fruits based on representatives of 48 olives from nine different publications [58]. With regard to pharmacokinetics, HT reached its maximum plasma concentration at 13 min (*t*_max_) following administration but was undetected after 1 hour. The approximate half-life of HT was 8 min, and the bioavailability of free HT ranges from 2.4 to 11.8% [59]. An in vivo toxicological evaluation reported that 500 mg/kg/d of HT did not show any mutagenic or genotoxic effects [60], with no genocytotoxity observed in vitro [61]. The promising safety profile thus made HT a potent nutraceutical supplement in the food and pharmaceutical industries. 

On the other hand, tyrosol (ty), another compound derived from oleuropein hydrolytic breakdown, is also an eminently researched antioxidant [62,63,64,65]. However, its 1-hydroxyl radical on the phenol ring makes it less effective in radical scavenging than hydroxytyrosol [66]. Nevertheless, tyrosol is still able to provide beneficial effects, as with other antioxidants, once it reaches its effective intracellular concentration [67,68]. The pharmacokinetics of tyrosol metabolites in rats were well studied by Lee et al., who demonstrated that the rapid uptake of tyrosol occurs within an hour. It is rapidly distributed to most organs before being eliminated within 4 h [69].

## 3. Respiratory Diseases

Respiratory diseases affect the airways, lungs and respiratory muscles surrounding the ribcage. These debilitating disorders can cause chest discomfort, wheezing, coughing and other respiratory symptoms [70]. Respiratory diseases, a significant public health concern, can be acute or chronic, and cause mild to severe reactions, with some being life-threatening. Infections, environmental pollutants, allergens, genetic predisposition and lifestyle factors such as smoking are among the causes of respiratory disease. The underlying cause may be due to viruses, bacteria, fungi or other microorganisms, leading to respiratory illnesses such as pneumonia, tuberculosis (TB) and even COVID-19 [71,72]. Meanwhile, environmental pollutants and allergens such as dust, pollen and smoke can trigger asthma and other respiratory illnesses in susceptible individuals [73]. Genetic factors can also contribute to respiratory diseases such as chronic obstructive pulmonary disease (COPD) and lung cancer [74]. Respiratory diseases can be classified according to their causes, symptoms and other characteristics. Table 3 summarises some standard respiratory disease classifications [70]. 

The most common respiratory diseases include asthma, pneumonia, tuberculosis, lung cancer, chronic obstructive pulmonary disease (COPD), pulmonary fibrosis, cystic fibrosis, sleep apnoea, allergic rhinitis and acute respiratory distress syndrome (ARDS).

Asthma is a chronic respiratory disease that affects the airways by causing the airways to become narrow and inflamed, with an increased production of mucus [75]. Asthma symptoms include wheezing, coughing, chest tightness and shortness of breath, making breathing difficult for individuals with asthma. Asthma is often triggered by exposure to allergens such as pollen, mould, pet dander, dust mites and certain foods, and its symptoms can vary depending on the individual and the origin of triggers [73]. Asthma attacks can occur suddenly and can be life-threatening if not treated promptly. Pneumonia is a lung infection caused by bacteria, viruses or fungi, which leads to symptoms such as cough, fever, chills, shortness of breath, chest pain and fatigue [76]. This disease typically spreads via respiratory droplets, such as coughing or sneezing. Pneumonia treatment depends on the infection’s underlying cause (a bacterial infection will be treated with antibiotics, etc.). Another example of respiratory disease caused by bacterial infection is tuberculosis (TB). Active TB disease occurs when the bacteria multiply and cause symptoms such as cough, fever, weight loss and night sweats, which can be fatal if left untreated [77]. In 2020–2021, an estimated 1.6 million people died from TB. Measures such as practicing good hygiene, vaccination and avoiding close contact with people who are sick are among good practices in preventing infections [71,72].

Allergic rhinitis is an allergic reaction to airborne substances such as pollen, dust mites, animal dander or mould. Upon a trigger by allergens, the T cells (predominantly T helper 2) will infiltrate the nasal mucosa and release cytokines that promote immunoglobulin E (IgE) production. This event will trigger the release of histamine and other mediators that cause inflammation and irritation of the nasal passages [78]. Symptoms of allergic rhinitis include sneezing, rhinorrhoea (runny nose), itching of the nose or throat, watery or red eyes and postnasal drip [79]. Treatment for allergic rhinitis includes avoiding the allergen triggers and using medications such as antihistamines, intranasal corticosteroids and decongestants to relieve the symptoms [78,79]. Chronic obstructive pulmonary disease (COPD) is a progressive lung disease typically caused by long-term exposure to irritants such as cigarette smoke, air pollution and dust [80,81]. The main symptoms of COPD include cough, shortness of breath, wheezing and chest tightness, which will reduce the patient’s quality of life [80]. However, as the disease progresses, it will lead to other complications such as respiratory infections, heart problems and lung cancer [81]. Treatment for COPD includes medications, such as bronchodilators and inhaled corticosteroids, which can help to open the airways and reduce lung inflammation. Nonetheless, the early diagnosis and treatment of COPD can help to slow its progression and improve one’s quality of life [80].

Pulmonary fibrosis is a chronic lung disease that occurs when lung tissue becomes damaged and scarred, leading to difficulties in the proper functioning of the lungs. Over time, the scarring can progress and lead to respiratory failure [82]. Pulmonary fibrosis symptoms include shortness of breath, cough and fatigue. Pulmonary fibrosis can be caused by various factors, including exposure to environmental irritants, radiation therapy, certain medications, autoimmune diseases, such as rheumatoid arthritis or lupus, and idiopathic pulmonary fibrosis (IPF) [83]. Treatment for pulmonary fibrosis typically focuses on managing the symptoms and slowing the progression of the disease. Medications such as corticosteroids and immunosuppressants can be used to reduce inflammation and scarring in the lungs; however, avoiding exposure to irritants, quitting smoking and good respiratory hygiene should be practiced [84].

Cystic fibrosis (CF) is a genetic disease that can affect the lungs. Mutations in the cystic fibrosis transmembrane conductance regulator (CFTR) gene lead to thick, sticky mucus that can clog the airways and cause infections [85]. Common symptoms of CF include chronic cough, recurrent lung infections, shortness of breath and wheezing. The treatment for CF involves a combination of medications, airway clearance techniques and nutritional support. Drugs such as antibiotics, bronchodilators and mucolytics can help to clear the airways and prevent infections [86]. While there is no cure for CF, early diagnosis and treatment can help to manage symptoms and improve one’s quality of life.

A more progressive respiratory disease is lung cancer. Lung cancer stems from the over-proliferation of lung cells and is among the leading causes of cancer-related deaths worldwide [87]. The two main types of lung cancer are small-cell lung cancer (SCLC) and non-small cell lung cancer (NSCLC) [88]. NSCLC is the most prevalent, making up about 85% of cases [89], and it normally advances more slowly than SCLC. Adenocarcinoma, squamous cell carcinoma and giant cell carcinoma are among the subtypes of NSCLC. Although it tends to spread and develop more quickly than NSCLC, SCLC is less common and makes up only about 10–15% of all lung cancer cases. It is also frequently more responsive to chemotherapy [90]. Exposure to radon, air pollution and other carcinogens can increase the risk of developing lung cancer in addition to smoking [88]. Lung cancer may be characterised by chest pain, wheezing, shortness of breath and weight loss.

Inflammation, oxidative stress and microbial infections are factors that contribute to the development and progression of respiratory diseases. Inflammation induces lung tissue damage and chronic diseases such as asthma and chronic obstructive pulmonary disease (COPD) [91]. Oxidative stress, on the other hand, contributes to the development of respiratory disorders in addition to the further deterioration of lung tissue. ROS cause damage to DNA, proteins and lipids in the respiratory tract, causing inflammation and tissue damage [92]. This oxidative stress has been linked to the development of respiratory diseases such as COPD, asthma and lung cancer [91,93]. Inflammation induced by microbial infections, such as those caused by bacteria or viruses, contributes to the development of respiratory diseases such as pneumonia and bronchitis [94]. All three factors interact to increase the severity of respiratory diseases and their symptoms. Understanding the intricate relationship between inflammation, oxidative stress and microbial infections can aid in developing new treatments and prevention strategies for respiratory disease. Natural compounds that are enriched with anti-inflammatory, antioxidant and infectious properties could possibly mitigate respiratory disease development. Henceforth, a focus on the olive compounds’ action in respiratory health will be uncovered. 

## 4. Olive Bioactive Molecules in Respiratory Inflammation

Inflammation is an essential defence mechanism of the respiratory system which is triggered by foreign pathogens or internal damage to host cells [95]. The airway epithelium is the first line of defence that produces mucins, lysozyme, nitric oxide, defensins and lactoferrin to protect the respiratory tract [96]. Microbial products (i.e., LPS, viral dsRNA and ssRNA) and cytokines (i.e., IL-1β, IL-6 and TNF-α) promote intracellular signalling activation which leads to the production of inflammatory mediators via the interaction with pattern recognition receptors (PRRs) of Toll-like receptors (TLRs) [97,98]. Epithelial cells are able to recruit inflammatory cells by secreting cytokines such as TNF-α, IL-1β, macrophage colony-stimulating factor (GM-CSF) and platelet activators during inflammation [99]. The expression of CD11b in alveolar macrophages is a novel biomarker in obstructive lung disease [100]. Macrophages in COPD patients were polarised towards a pro-inflammatory M1 phenotype, which was associated with increased levels of inflammatory cytokines in the lungs [91,101]. Cigarette smoke also increased the CD11b^+^ pulmonary macrophages, including monocyte-derived alveolar macrophages [98], and activated macrophage polarisation [102]. Bigagli et al. reported that Hydroxytyrosol inhibited oxidative burst and CD11b expression of human granulocytes and monocytes [103], which potentially could improve obstructive lung diseases too. 

On the other hand, biomarkers in respiratory inflammatory diseases can be categorised by (1) inflammatory cells count (e.g., total white blood cells, PMN count); (2) cytokines and chemokines (e.g., TNF-α, IL-6, IL-8); (3) adhesion molecules (e.g., P-selections, VCAM, ICAM-1); (4) inflammatory proteins (e.g., C-reactive protein (CRP), serum amyloid A); (5) inflammatory enzymes (e.g., MMP-9, SOD, iNOS and COX-2); (6) enzymatic antioxidants (e.g., CAT, GPx, SOD) and non-enzymatic antioxidants (e.g., GSH, NPSH, ascorbic acid) and (7) oxidative stress products (e.g., ROS, RNS, MDA, AOPP, peroxynitrite) [98,104,105,106,107]. Luteolin, a flavone from olives, could inhibit ICAM-1 expression, IkappaB kinase (IKK) and nuclear factor-kappab (NF-kb) in respiratory epithelial cells. Luteolin and apigenin suppress mitogen-activated protein kinase (MAPK) pathways [108]. Similarly, essential oil from Abies holophylla leaf inhibits MAPK and NF-kb transcriptional activity, thus salvaging airway inflammation and epithelial hyperplasia [109]. C-reactive protein (CRP) is released during inflammatory events; therefore, CRP levels correlate with disease severity, lung function, volume alterations and pneumonia development [105,110]. A double-blind controlled trial involved 93 ventilated pulmonary failure patients given two high-fat diets, one with olive, and the other with sunflower oil. The olive oil supplementation group had lower CRP levels and increased antioxidants in serum, while the sunflower oil group did not [111]. Interestingly, olive phytochemicals such as oleuropein and oleocanthal showed better interaction with CRP than ibuprofen in molecular docking, which can be exploited as an analgesic and anti-inflammatory drug [112]. 

In addition, respiratory viral infections such as SARS-CoV-2 trigger a substantial production of chemokines such as IL-6, IL-8 and IL-1β. COVID-19 patients’ blood profiles have been detected with a high level of IL-6 [113]. Studies on COVID-19 have reported that the mortality of patients correlates with a “cytokine storm” [114]. A cytokine storm creates excessive pro-inflammatory cytokines, resulting in tissue damage, multi-organ failure and eventually death [115]. Multiple drugs are utilised to suppress extreme cytokine activity [116]. COVID-19 patients are prescribed cytokine blockers to inhibit cytokine synthesis. In a case report of a 42-year-old COVID-19 male patient, Tocilizumab (TCZ), an IL-6 inhibitor, improved his recovery [117]. Thus, natural compounds that are enriched with anti-inflammatory effects could suppress excessive cytokine production. 

4-Hydroxyphenylacetic acid (4-HPA) from olives was observed to inhibit the expression of TNF-α, IL-1β and IL-6 in lung-injured rat tissue [118]. It also decreased hypoxia-triggered host hypoxia-inducible factor-1α (HIF-1α) in alveolar epithelial cells (AECs), where HIF-1α was also found to be a critical factor in promoting SARS-CoV-2 infection and subsequent COVID-19 pro-inflammatory responses [119]. Furthermore, 4-HPA suppressed lung oedema and inflammation in hypoxia-induced rat models. Similarly, luteolin decreased lung oedema in a septic experimental mouse model. An intraperitoneal injection of luteolin significantly decreased pro-inflammatory cytokines IL-6 and IL-1β in lung tissue [120]. It further impedes the expression of ICAM-1 and NF-kb. Neutrophilia was reported to occur in 8 to 10 patients after COVID-19 symptom onset and was highly correlated with lung injury in COVID-19 patients [121]. Oleuropein has also been reported to attenuate neutrophil infiltration by lowering myeloperoxidase (MPO) levels in male mice [122]. The unsaponifiable fraction of extra virgin olive oil (EVOO) attenuated DNA damage in an oxidative-stress-induced mice model. It improved lung histoarchitecture and became a new supplementation strategy to reduce lung inflammation [123]. Similarly, tyrosol has been demonstrated to inhibit pro-inflammatory cytokines such as TNF-α, IL-1β, IL-6 and Cyclooxygenase (COX-2) protein expression in LPS-induced lung inflammation [124]. COX-2 is essential in producing PGE_2_, which is responsible for pain expression and inflammation [125]. A perfect example is the SARS coronavirus outbreak in 2003, whereby the elevation of PGE production was reported because of direct virus binding to the COX-2 promoter [126]. Lescure et al. proposed that PGE_2_ plays a crucial role in COVID-19 pathophysiology, hyperinflammatory and immune responses. He hypothesised that PGE_2_ inhibition can increase the host’s immune response against COVID-19 [127].

Along with tyrosol and hydroxytyrosol, oleuropein was also reported to attenuate COX-2 expression in animal and human models [128]. Maslinic acid from olives was also found to downregulate the production of COX-2/PGE_2_ in lipopolysaccharide (LPS)-induced cells [129]. Likewise, Hydroxytyrosol (HT) and oleuropein (OLE) reduced COX-2 expression in a TNF-a-induced pre-senescent human lung fibroblast. Interestingly, these olive derivatives protected the cells from senescence by suppressing senescence-associated secretory phenotype (SASP) markers [128]. A molecular docking study by Thangavel et al. with OliveNet™, a directory of *Olea Europaea* (Oleaceae), found that six different olive secoiridoids could prevent the hyperinflammatory responses of SARS-CoV-2 [130]. Similarly, olive oil consumption is linked to preventing non-communicable diseases and COVID-19, as seen via a detailed search of papers published in the last 30 years [131].

## 5. Olive Bioactive Molecules in Respiratory Oxidative Stress

Oxidative stress and inflammation are significant events, particularly in chronic obstructive pulmonary disease (COPD) patients [132] and in COVID-19 pathogenesis [133]. The hyperproduction of neutrophils is also responsible for ROS production, which induces oxidative stress and ultimately affects the lung defence system [134]. Derouiche et al. analysed the importance of antioxidant therapeutics to counteract the severity of lung diseases associated with SARS-Cov-2 (COVID-19) via a systematic review [133]. Comparably, Lammi et al. suggested taking on food-derived antioxidants as a new strategy to treat COVID-19 patients to decrease oxidative stress in the respiratory system [135]. Our review reinforced the idea of using antioxidant therapy by collecting experimental findings from related articles. A hydrophilic fraction of extra virgin olive oil (OOHF) diminished lung oxidative stress caused by aluminium and acrylamide. OOHF decreased malondialdehyde (MDA), hydrogen peroxide H_2_O_2_ and advanced oxidation protein product (AOPP), and improved the lung histoarchitecture [123]. 

During acute respiratory distress syndrome (ARDS), the lungs generate inflammatory factors which subsequently increase the production of inducible nitric oxide synthase (iNOS) via the neutrophil and bronchial epithelium. Higher iNOS availability substantially increases NO production in the lung tissue [136,137]. Abundant NO metabolites cause necrosis and pulmonary epithelial cell denaturation in ARDS patients [138,139]. Therefore, bioactive compounds or drugs that could reduce iNOS expression could subsequently be utilised to subside the inflammatory reactions. Tyrosol efficiently reduced iNOS and NO expression in LPS-stimulated macrophages [124] and an LPS-induced acute lung injury (ALI) mouse model by inhibiting NF-κb and activator protein-1 (AP-1) [124]. In parallel, maslinic acid (MA) also downregulated iNOS in the lung tissue of LPS-treated mice via suppressing NF-κB and p-STAT expression [129]. Additionally, Oleuropein aglycone also improved carrageenan-induced pleurisy by inhibiting inflammatory cytokines, NO and lipid peroxidation [122]. 

Transcription factor nuclear factor erythroid 2-related factor 2 (Nrf2) is an emerging regulator of antioxidant defence. Nrf2 has also been well reviewed in preventing respiratory diseases such as ARDS, COPD, asthma and lung cancer [140]. Most of the olive antioxidants, including MA [129], tyrosol [141] and olive extract [141], activated the Nrf2/HO-1 pathways to prevent oxidative stress. Clear details about the studies conducted on olive benefits in the respiratory system are tabulated in Table 4. The majority of in vivo studies were conducted with regard to lung edema, pneumonia, lung injury and asthma animal models. Olive compounds reversed the clinical manifestations of the diseases mainly by exhibiting antioxidant pathways and inhibiting pro-inflammatory and cytokine production. Additionally, research is also conducted in both in vitro and in vivo models where respiratory epithelial cells, fibroblasts and macrophages represent in vitro models. Olive compounds regulate biomolecules and pathways in respiratory inflammation and oxidative stress, as seen via in vitro studies, illustrated in Figure 2, while Figure 3 represents olive compounds’ bioactivity in vivo. 

A randomised clinical trial in China studied the effect of olive-oil-based (OLIVE) and soybean-oil-based (SOYBEAN) parenteral nutrition. The olive is demonstrated to deliver adequate nutrients and is well tolerated in the system. The oxidation and inflammation effects are similar for both the olive and soybean groups. Interestingly, olive-oil-based nutrition groups were found to significantly lower infection incidence. These outcomes signify the immune-boosting capability of the olive-based parenteral nutrition [146]. In addition, the olive extract supplement significantly reduced the sick days of high school athletes suffering from upper respiratory illness. However, there was no significant difference between olive extract intake and the incidence of illness [147]. A high-protein diet including olive oil reduced the arterial CO_2_ tension serum hs-CRP level in acute pulmonary failure patients [111]. Table 5 shows clinical trials based on olive bioactive compounds in respiratory disease patients, while Table 6 presents unpublished clinical trials. 

## 6. Olive Bioactive Molecules in Infectious Respiratory Diseases

Respiratory infections occur due to bacteria or invading viruses. Antibiotic treatments tend to fail when dealing with antibiotic-resistant bacteria. Respiratory pathogens were reported to exacerbate chronic obstructive pulmonary disease [154]. A meta-analysis of 3338 COVID-19 patients reported that 6.9% of patients were coinfected with bacterial infections [155]. Seven compounds from olive (caffeic acid, verbascoside, oleuropein, luteolin 7-*O*-glucoside, rutin, apigenin 7-*O*-glucoside and luteolin 4’-*O*-glucoside) were found to have an antibacterial effect towards strains such as *Bacillus cereus*, *Staphylococcus aureus*, *Pseudomonas aeruginosa* and *Klebsiella pneumoniae* and antifungal strains such as *Candida albicans* [156]. Olive secoiridoides also inhibited five different bacterial strains (*Haemophilus influenzae*, *Moraxella catarrhalis*, *Salmonella typhi*, *Vibrio parahaemolyticus* and *Staphylococcus aureus*) that commonly cause intestinal and respiratory tract infections [157]. Moreover, aliphatic aldehydes from olives showed similar antibacterial activity [158,159] where alpha- and beta-unsaturated aldehydes were found to have broad-spectrum antibacterial activity, while saturated aldehydes did not show a significant antibacterial effect. Olive extract was reported as being one of the most potent antimycobacterial agents among 63 Mexican traditional medicines postulated as a potential drug for tuberculosis [160,161]. 

Viral infection is the major reason for respiratory diseases such as pneumonia, bronchitis and COVID-19. The most common viruses that invade the human respiratory system are human coronavirus, rhinovirus (RV), influenza, adenovirus, respiratory syncytial virus (RSV) and so on [162]. Olive compounds have been well reviewed and preferred as a functional food containing antiviral and immune-boosting effects [163]. Additionally, Hydroxytyrosol has been found to disrupt the viral envelope of influenza A viruses, including H1N1, H3N2, H5N1 and H9N2 [164]. Oleuropein has also been reported to inhibit the herpes simplex virus (HSV-1) via phosphorylating PKR, c-FOS and c-JUN in Hela cells [163]. Furthermore, purified HT from olive and a patented HT, HIDROX^®^, have been shown to inactivate SARS-CoV-2. They altered the spike protein, significantly impacting the viral genome [165]. A similar effect has been reported in molecular docking by Geromichalou et al. He demonstrated the EVOO compound’s potential to bind to inhibit the SARS-CoV-2 spike protein via targeting angiotensin-converting enzyme 2 (ACE2) and the receptor-binding domain (RBD) [166]. On the other hand, Nrf2 has also been revealed to have the ability to inhibit virus penetration by secreting anti-proteases in COVID-19 patients [167]. Nrf2 activates interferon gene expression to initiate antiviral activity [168]. Our review collectively reported findings of olive-derived phytochemicals’ ability to activate the Nrf2 pathway [129,141,142]. Thus, they certainly could play a role in drug design for COVID-19 treatment. 

## 7. Olive Bioactive Molecules in Over-Proliferation of Respiratory Cells

Lung cancer is a leading cause of cancer death to date, which is due to the over-proliferation of respiratory cells [169]. Olive compounds, especially polyphenols, have been well studied for their anticancer effects [44,170,171]. A systematic search and meta-analysis of 45 studies [22] discovered that olive oil consumption prevents cancer. An olive extract and bromelain combination suppressed Benzo[a]pyrene (BaP)-induced lung carcinogenesis by decreasing the expression of inflammation and oxidative markers (Nrf2, NF-κB) [142]. In another study, oleic acid, and its metabolite oleoyl ethanolamide, induced apoptosis in lung carcinoma cell lines by decreasing programmed death-ligand 1 (PD-L1), the tumorigenesis marker and the phosphorylate STAT pathway [172]. An extract from olive mill wastewater (OMWW A009) limited lung cancer cell propagation by activating apoptosis. The extract was able to reduce CXCL12 and CXCR4 chemokines and STAT3 phosphorylation [173]. Besides, (-)-Oleocanthal (OC) disrupted metastasis by inhibiting the activation of mesenchymal-epithelial transition factor (c-MET) and cyclooxygenase 2 (COX2) in adenocarcinoma cells A549 and NCI-H322M [174]. The same study showed that eight weeks of OC supplementation prevented brain and other organ metastasis in mice models. The c-MET inhibitors showed promising results in lung cancer prevention in both animal models and clinical trials [175,176]. Hydroxytyrosol was also reported to have reversed TGFβ1-induced EMT in respiratory epithelial cells by inhibiting AKT and SMAD2/3 expression [177]. Thus, hydroxytyrosol could be exploited for cancer prevention by targeting c-MET inhibition. 

## 8. Conclusions and Future Perspectives

This review has highlighted several beneficial effects of olive compounds in respiratory diseases. In vivo and in vitro studies and clinical trials have shown promising results. Most of the in vivo and in vitro studies have been conducted in single olive compounds, which is good for elucidating the potential of olive compounds individually. Nevertheless, combined olive polyphenols’ synergic mechanism could accelerate human health. Hence, studies with the combination of different polyphenols will be valuable. Aside from this, the bioaccessibility and bioavailability of the compounds have been neglected in recent study designs, as tabulated in Table 4. Bioavailability varies depending on the chemical structure, purity of compounds, animal models, etc. Therefore, researchers should consider evaluating the bioaccessibility and bioavailability to provide a better understanding of these compounds in the body. There is an apparent gap in the epigenetic regulation of olives in respiratory models. More studies should focus on the elucidation of an olive effect, especially in DNA methylation and histone modification, etc. 

To conclude, our review summarised the significance of olive-derived phytochemicals in ameliorating lung diseases. In vitro, in vivo and clinical trials support the notion that olive-derived phytochemicals exert respiratory protection mainly by subsiding the production of pro-inflammatory cytokines and oxidative stress, therefore potentially abating a cytokine storm in COVID-19 patients. The nutraceutical product derived from olives is highly recommended as it provides a protective in vitro and in vivo environment in reducing the incidence of respiratory diseases associated with chronic inflammation and oxidative stress, as summarised in Figure 4. 

## Figures and Tables

**Figure 1 antioxidants-12-01140-f001:**
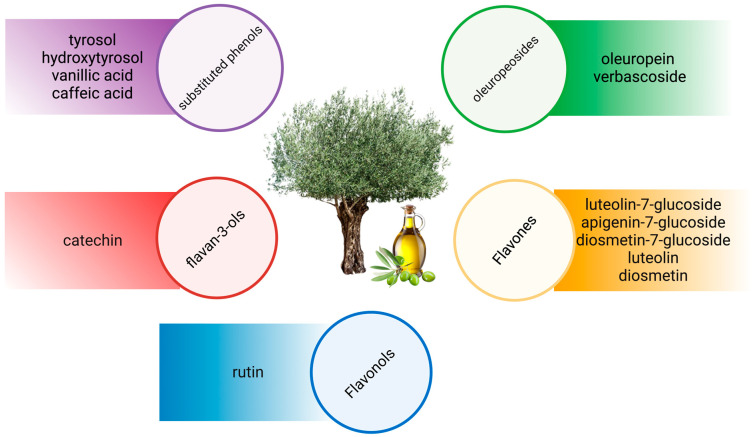
Phenolic compounds in the olive tree.

**Figure 2 antioxidants-12-01140-f002:**
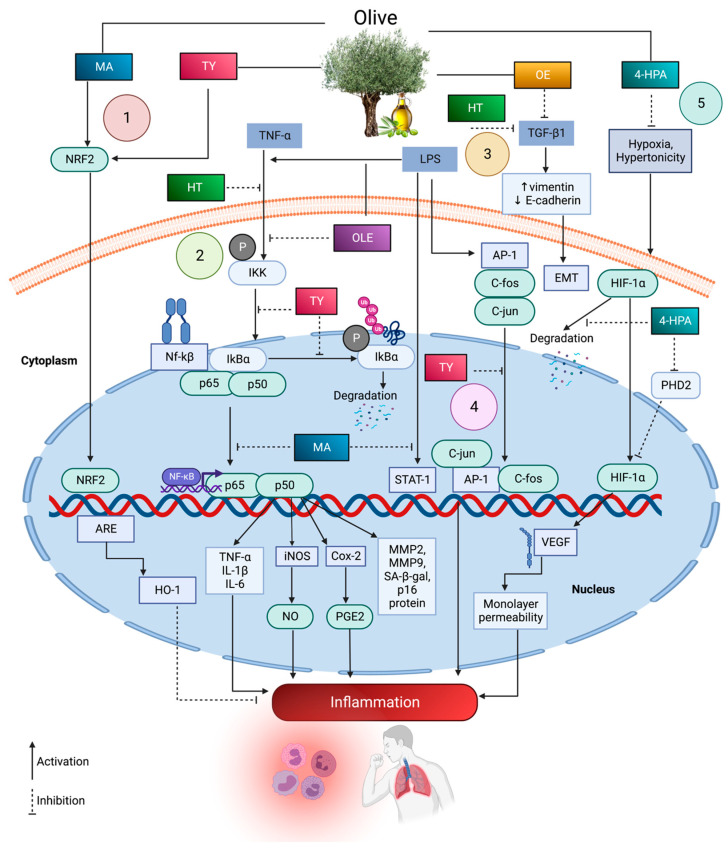
Summary of molecular pathways regulated by olive phytocompounds that modulate respiratory disease inflammation against stimuli in vitro. (1) Upregulation of Nrf2/OH-1 antioxidant pathways by MA and Ty. (2) HT, TY and OLE attenuate TNF-α-induced inflammation and downregulation of NF-κb pathway. LPS or TNF-α activates IKKs and triggers the phosphorylation of inactive IκBα-NF-κb complex. Ty, OLE, HT and MA hinder the ubiquitylation and degradation of phosphorylated IκBα along with the prevention of NF-κb, p50 dimer nuclear translocation and the further activation of target genes for chemokines, cytokines, adhesion molecules, Cox-2/prostaglandin, iNOS/nitric oxide and senescence-associated markers. (3) EMT reversion via the olive extract. (4) Downregulation of AP-1 pathway by attenuating LPS-induced overexpression of c-fos/c-jun dimer by TY. (5) Downregulation of HIF-1 pathway. 4-HPA ameliorates hypoxia and hypertonicity-induced hypoxia-inducible factors; 1 alpha activation by promoting pyruvate dehydrogenase and two dependent hypoxia-inducible factors; 1 alpha degradation and inhibits subsequent vascular endothelial growth factor synthesis and a further increase in monolayer permeability.

**Figure 3 antioxidants-12-01140-f003:**
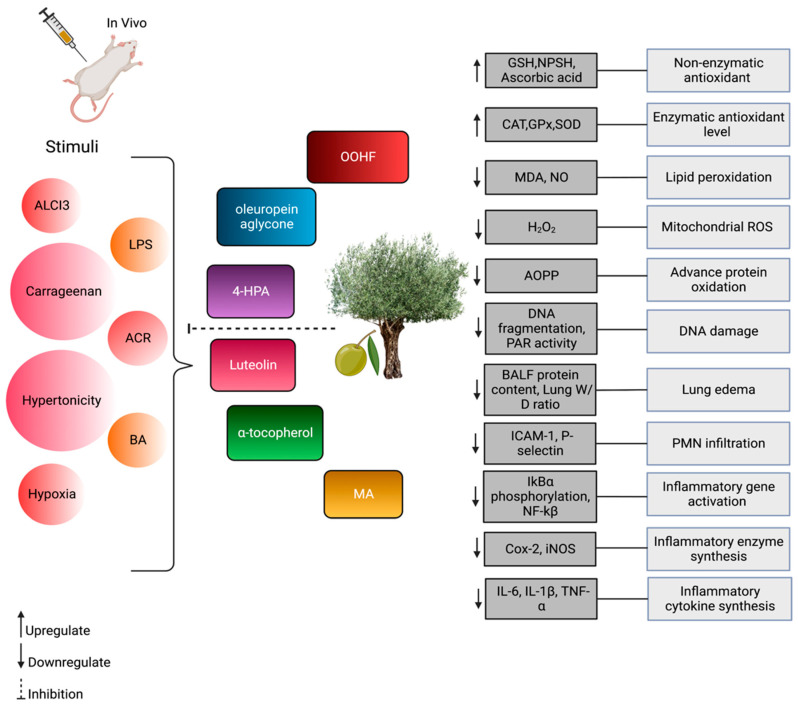
Representation of olive and its phytoconstituent bioactivity in vivo. Olive prevents the after-effects of stimuli. Olive attenuates several pathways and biomarker changes (dark grey) and prevents subsequent conditions (light grey).

**Figure 4 antioxidants-12-01140-f004:**
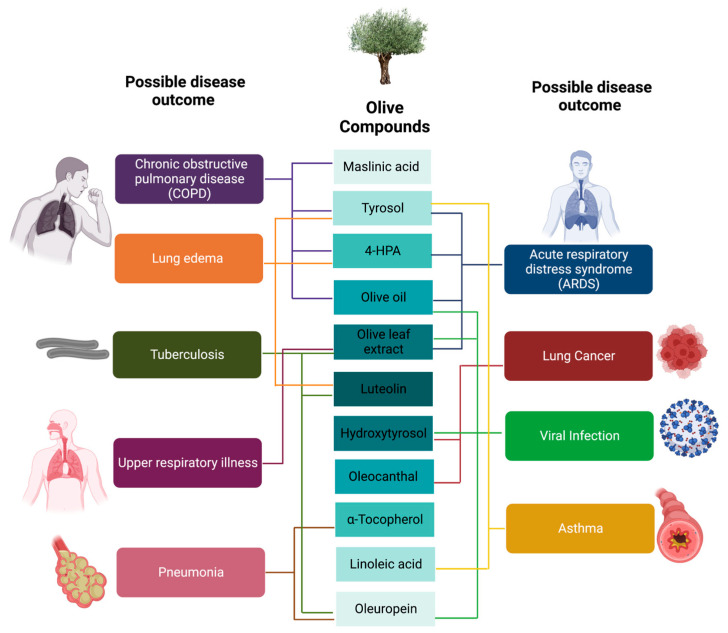
Summary of olive compounds and possible disease outcomes.

**Table 1 antioxidants-12-01140-t001:** Composition of olive oil.

Saponifiable Fraction	Unsaponifiable Fraction
Monosaturated fatty acids, e.g., oleic acid Polyunsaturated fatty acids, e.g., linolenic acid, palmitoleic acid Saturated fatty acids, e.g., stearic acid, myristic acid, palmitic acid	Hydrocarbon, e.g., lycopene, squalene and β-Carotene Volatile compounds Pigments, e.g., chlorophylls Hydrophilic phenolics, e.g., phenolic alcohol, phenolic acid, lignans and flavones Lipophilic phenolics, e.g., tocopherols and tocotrienols Sterols, e.g., alcohols, campesterol, β-sitosterol, aliphatic alcohols Non-glyceride esters, e.g., alcoholic and sterol compounds, waxes

**Table 2 antioxidants-12-01140-t002:** Main phenolic compounds in virgin olive oil [39].

Phenolic Acids	Phenolic Alcohols	Flavonoids	Secoroidoids	Lignans	Tocopherols and Tocotrienols
Gallic acid Vanillic acid Benzoic acid Cinnamic acid Caffeic acid Ferulic acid Coumaric acid	Hydroxytyrosol Tyrosol	Luteolin Apigenin Methoxyluteolin Rutin Anthocyanins Cyanidin	Oleuropein Oleocanthal Oleacein Ligstroside Ligstroside aglycon Nu¨zhenide	(+)-1-pinoresinol Phenylpropane (+)-1-acetoxypinoresinol	(α, β, γ, δ)

**Table 3 antioxidants-12-01140-t003:** Categories of respiratory diseases adapted from Kritek et al., 2018 [70].

Category	Examples
Infections Disease caused by various microorganisms, such as bacteria, viruses, fungi and parasites.	Sinusitis, pneumonia, tuberculosis, influenza and COVID-19
Obstructive Disease caused by narrowing or obstruction of the airways, causing difficulty in breathing.	Asthma, chronic obstructive pulmonary disease (COPD), bronchiectasis and bronchiolitis
Restrictive Disease affects the lung tissue or chest wall making it difficult for the lungs to expand and contract properly.	COPD, pulmonary fibrosis and chest wall disorders
Occupational Disease caused by exposure to various substances in the workplace, such as dust, chemicals and fumes.	Occupational asthma and pneumoconiosis
Genetic Disease caused by genetic mutations or abnormalities.	Cystic fibrosis and alpha-1 antitrypsin deficiency
Malignancy Disease caused by the uncontrolled growth of abnormal cells in the lungs or airways.	Lung cancer and mesothelioma
Vascular Disease affects the blood vessels in the lungs.	Pulmonary hypertension and pulmonary embolism

**Table 4 antioxidants-12-01140-t004:** Studies conducted on olive compounds in terms of respiratory anti-inflammation and antioxidants.

Compound	Model	Dosage and Duration	Finding(s)	Possible Contribution in Respiratory Disease	Reference
4-Hydroxyphenylacetic acid (4-HPA)	Rat alveolar epithelial cells (AECs) Seawater-aspiration-induced lung-injured rat	100 μg/mL— 4 h—in vitro 100 mg/kg 14 h—in vivo	↑ PHD2 protein level ↑ HIF-1α protein degradation ↓ Hypertonicity and hypoxia-induced HIF-1α protein IN AEC ↓ TNF-α, IL-1β and IL-6 4-HPA ↓ Hypoxia-derived lung tissue damage, vascular leakage, alveolar damage and lung edema ↓ Inflammatory cytokine expression (TNF-α, IL-1β, IL-6) ↓ BALF leukocyte number and inflammatory cell infiltration ↓ Hypoxia-induced HIF-1α in rats	Anti-inflammation	[118]
Luteolin	Caecal ligation and puncture (CLP)-induced ALI mice	0.2 mg/kg 1 h	↓ Lung edema, the water content in W/D weight ratio ↓ LPS-induced release of IL-6, IL-1β and plasma IL-6 and TNF-α cytokines ↓ mRNA expression of ICAM-1, PMNs infiltration ↓ NF-kB, iNOS ↓ MDA and ↑ CAT, SOD, GSH Restored in lung architecture	Antioxidant Anti-inflammation	[120]
Oleuropein aglycone	Carrageenan-induced pleurisy mice	40–100 mM/kg 30 min	↓ Degree of lung injury ↓ Neutrophil infiltration ↓ MPO, ICAM-1 and P-selectin ↓ Inflammatory cytokine TNF-α and IL-1β ↓ Lipid peroxidation with ↓ in NO, MDA, (ONOO^-^) RNS	Anti-inflammation	[122]
Hydrophilic fraction from olive oil (OOHF)	Aluminium- and acrylamide-induced rats	(1 mL) by gavage—21 days	↑ Final body weight ↓ AlCl_3_+ACR-induced lung weight loss ↓ (MDA), H_2_O_2_ and AOPP; thus, ↓ in lipid peroxidation ↓ Hemosiderin-laden macrophage number in the lungs	Antioxidant	[123]
Tyrosol	RAW 264.7 macrophages Lipopolysaccharide (LPS)-induced ALI mice	1 nM–10 mM 24 h—in vitro 0.1, 1 and 10 mg/kg) 1 h—in vivo	↓ LPS-induced NO ↓ LPS-induced degradation of IκBα ↓ NF-κB, pro-inflammatory cytokines (TNF-α, IL-1β, IL-6) iNOS, COX-2 ↓LPS-induced pathological changes in the lung tissue ↓ LPS-induced lung vascular leakage, lung edema ↓ LPS-induced inflammatory cell infiltration ↑ SOD antioxidant enzyme activity	Anti-inflammation	[124]
Maslinic acid	Human umbilical cord endothelial cells LPS-injected lung-injured mice	2–20 μM— 6 h—in vitro 0.07–0.7 mg/kg 1 day—in vivo	↓ LPS-induced TNF-α synthesis in BALF ↓ LPS-induced iNOS synthesis in lung tissue ↓ Pulmonary injury and histopathological injury score	Antioxidant Anti-inflammation	[129]
Oleuropein (OLE) Hydroxytyrosol (HT)	Pre-senescent human lung cells (MRC5)	OLE—10 mM HT—10 mM 4–6 weeks	↑ Cell proliferation ↓ Positive SA-βgal, p16 protein, IL-6 and MMPs ↓ NF-kB, COX-2 and α-SMA OLE ↓ reduces both the nuclearity and cell size of NHDF cells	Anti-inflammation	[128]
Ethanolic olive leaf extract (EOLE) and bromelain	Benzo(a)pyrene-mediated lung cancer mice	10, 20, 50, 100 and 200 mg/kg— phases 1 and 2—8 days, 3rd phase—16 weeks	↓ MDA, LDH, LPO, ROS ↑ GSH ↓ Lung epithelium necrosis ↓ BaP-induced lung toxicity ↓ TNFα, IL-6 ↓ MMP-2, MMP-9 ↑GPx, GR, GST, SOD and CAT ↑ Nrf2 ↓ NF-kb	Anticancer Antioxidant Anti-inflammatory	[142]
α-tocopherol (AT)	Aspiration-pneumonitis-induced rats	20 mg/kg/day, 7 days	↓ MDA ↑ CAT and SOD activity and ↓oxidative damage ↓ CC-16 synthesis ↓ In peribronchial inflammatory cell infiltration (PICI), alveolar septal infiltration (ASI), alveolar edema (AED), alveolar exudate (AEx), alveolar histiocytes (AH) and necrosis	Antioxidant	[143]
Oleuropein	Lipopolysaccharide (LPS)-induced ALI in rats	200 mg/kg 20 days	↑ GSH, GSH.Px and CAT ↓ MDA level of tissue and serum ↓ IL-6, MPO, NF-kB and TNF-α Improved acute interstitial pneumonia (AIP) histopathological changes induced by LPS	Antioxidant Anti-inflammatory	[144]
Tyrosol	Ovalbumin-induced asthma rats	20 mg/kg— once a day for 22 days	↓ MDA ↑ GSH and GPx ↓ NF-κB, TNF-α, IL-4, IL-5, IL-13, IFN-γ, IgE, ↑IL-10 ↓Mucus hypersecretion ↓ Eosinophils ameliorated OVA-induced histopathological lesions	Antioxidant Anti-inflammatory	[145]

↑ = Increases/upregulations, ↓ = decreases/downregulations. PHD2, HIF-prolyl hydroxylase domain; HIF-1α, Hypoxia-inducible factor-1α; TNF-α, tumour necrosis factor α; IL-1β, interleukin-1β; IL-6, interleukin-6; ICAM-1, intercellular adhesion molecule-1; NF-κβ, nuclear factor kappa-light-chain-enhancer of activated B cells; polymorphonuclear leukocyte (PMNs); iNOS, inducible nitric oxide synthase; MDA, malondialdehyde; CAT, catalase; SOD, Superoxide dismutase; GSH, glutathione; MPO, Myeloperoxidase; AlCl_3_, aluminium chloride; ACR, acrylamide; H_2_O_2_, hydrogen peroxide; AOPP, advanced oxidation protein products; LPS, lipopolysaccharide; IκBα, nuclear factor of kappa light polypeptide gene enhancer in B-cell inhibitor, alpha; LPO, lipid peroxidation; HO)-1, heme oxygenase; (Nrf2), nuclear factor erythrocyte 2-related factor 2; GPx, Glutathione peroxidase; ROS, reactive oxygen species; COX-2, cyclooxygenase; PMNs, polymorphonuclear neutrophils; prostaglandin, PGE2; TGFβ1, transforming growth factor; LPS, lipopolysaccharide; AREs, antioxidant response elements; MMP, matrix metalloproteinase; SASP, senescence-associated secretory phenotype; α-SMA, alpha smooth muscle actin; AP-1, activator protein; BALF, bronchoalveolar lavage fluid; NAG, N-acetyl glucosaminidase; LDH, lactate dehydrogenase; GGT, γ-glutamyl transferase; PICI, Peribronchial inflammatory cell infiltration; ASI, alveolar septal infiltration; AED, alveolar edema; AEx, alveolar exudate; CC-16, Clara cell protein 16; STAT-1, signal transducer and activator of transcription.

**Table 5 antioxidants-12-01140-t005:** Clinical trials based on olive bioactive compounds in respiratory diseases.

ID	Intervention and Control	Disease/ Condition	Study Size	Finding	References
NCT02421614	Control group High-protein diet including protein: 20%, fat: 30% (sunflower oil) Group A: Olive and sunflower oil 45%, carbohydrate: 35% (https://www.sciencedirect.com/topics/nursing-and-health-professions/sunflower-oil, accessed on 26 March 2023) Group B: 20%, sunflower oil: 45%, carbohydrate: 35% ** Dose and duration ** 8 times per day for 14 days	Acute respiratory failure	48 ventilated acute pulmonary failure patients	↓ Arterial CO_2_ tension ↓ Serum hs-CRP level ↑ Serum antioxidant capacity Sunflower oil does not show any beneficial effects	[111]
NCT01674595	AVANZ^®^ Olive ** Dose and duration ** 300, 600, 3000, 6000 and 15000 SQ+weekly for 6 weeks	Allergic rhinoconjunctivitis	93 allergic rhinoconjunctivitis with/without asthma	All adverse drug reactions are mild and not serious All patients fully recovered ↑ IgG4 and IgE levels ↓ Immediate skin reactivity	[148]
NCT00876356	Yogurt-conjugated linoleic acid (CLA), 3 g CLA (75% c9,t11-CLA)—87% purity Placebo—safflower oil ** Dose and duration ** One daily portion for 12 weeks	Asthma	29 asthmatic children	Pulmonary function is the same as the placebo Both groups improved airflow IFN-γ and IL-4 remained unchanged No difference in plasma eosinophilic cationic protein in both groups ↑ Urinary 8-oxodG in both groups	[149]
ACTRN12618000328279	Group A: OLE (to 20 g of olive leaf, containing 100 mg oleuropein) Group B: Placebo ** Dose and duration ** 1 tablet daily for 2 months	Upper respiratory illness	32 high school athletes	↑ Urinary 8-oxodG in both groups ↓ Occurrence of sick days by 28% ↓ Illness days in females ↑ Increase in illness day in males 4 out of 5 participants reported adverse effects with symptoms of stomach aches, headaches, bad skin and acne	[147]
	Group A: OliClinomel N4—olive-oil-based parental nutrition (OOPN) Group B: Soybean-based parental nutrition	N/A	226 patients undergoing surgery	OOPN ↑serum albumin level OOPN ↑ prealbumin levels compared to soybean-oil-based parental nutrition (SOPN) OOPN ↑ serum IGF-I levels on 14th day OOPN has greater ↑ in serum oleic acid level compared to SOPN OOPN ↓ IL-6 levels compared to control OOPN ↓ occurrence of lung infection compared to SOPN OOPN ↓ experiences an infection or infestation compared to SOPN	[146]

**Table 6 antioxidants-12-01140-t006:** Unpublished clinical trials based on olive bioactive compounds in respiratory diseases.

ClinicalTrials.gov ID	Intervention and Control	Disease/Condition	Study Size	Findings/Outcome Measures	Status	References
NCT05685901	High polyphenolic olive oil (early harvest olive oil)—2 mL ** Dose and duration ** Twice a day for 30 days	COVID-19 prevention	88 adults	COVID-19 incidence Resolution time of symptoms Differences in severity of symptoms	Completed Results not published	[150]
NCT04873349	Nusapure standardised olive leaf capsule, 750 mg (50% oleuropein) Placebo: starch ** Dose and duration ** Twice a day for 10 days	COVID-19 respiratory infection COVID-19 acute bronchitis	60 COVID-19 adults	Clinical symptoms Viral clearance Mortality rate Improvement in analysis (CBC, CRP, LDH, ESR, Ferritin, D-dimer, creatinine, ALT and AST)	Completed Results not published	[151]
NCT01734265	Group A: Depigoid 50% Grasses 50% Olea europaea (2000DPP/mL) Group B: Depigoid 50% Grasses 50% Parietaria judaica (2000 DPP/mL)—0.2 mL followed by 0.3 mL after 30 min if no adverse events occur ** Dose and duration ** Second maintenance dose of 0.5 mL up to 4 weeks	Allergy Rhiniti Rhinoconjunctivitis Seasonal asthma	63 adults	Number of subjects (%) suffering immediate and/or delayed local reactions Immediate and/or delayed local reactions Immunology assessment: sIgE and sIgG4	Completed Results not published	[152]
NCT01096771	Group A: ClinOleic 20% Group B: Intralipid 20% ** Dose and duration ** 96 h continuous infusion	Acute respiratory distress syndrome	14	Bronchoalveolar Lavage Fluid Interleukin-8 Concentrations Ventilator days PaO2:FiO2 ratio Infection rate Biomarkers Organ failure	Terminated due to low recruitment rate	[153]

## Data Availability

Not applicable.

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
