# Peer review of "Role of Olive Bioactive Compounds in Respiratory Diseases"

_antioxidants, 2023, doi:10.3390/antiox12061140_

Round 1

Reviewer 1 Report

In this manuscript, the authors tried to provide insight into the therapeutic potential of olive bioactive compound’s antioxidant, anti-inflammatory and antiviral properties in the treatment of respiratory diseases. 

In Section 2 "Olive bioactive compounds", several active components in olive oil or olive plants were mentioned. However, it was unclear that any of those compounds have been tested as a potential treatment in respiratory diseases. 

The evidence of the beneficial effects of olive bioactive compounds on respiratory diseases was not conclusive and convincing. For example, in Section 4 "Olive compounds in respiratory inflammation", most of studies cited were in vitro studies. There were only one clinical study on olive oil-based diet and one preclicial study of the effect of 4-HPA in a rat model of lung injury. 

Were there any clinical studies on the effect of individual olive bioactive compounds on any respiratory disease? 

Reviewer 2 Report

In the present review Vijakumaran and colleagues focused on the healthy potential of olive bioactive molecule to cope with the adverse effects leading to respiratory diseases, such as COPD, asthma, infectious diseases and lung cancer. The authors aimed to underling the main evidence supporting a potential role of olive bioactive molecules in respiratory system protection against inflammation, oxidative stress and infection.

Although the topic is of interest, many issues need to be properly addressed before considering the manuscript suitable for publication.

Generally speaking, the field of respiratory disease is very wide as well as the classes of bioactive phytochemicals in olive tree, leaves and fruit. In my opinion these topics must be reviewed more in detail. The authors should provide a schematic and more compressive analysis of the “fil rouge” connecting the disease outcomes and the healthy potential of olive bioactive molecules. Otherwise, I suggest to focus on a few diseases, choosing those etiologically similar, as well as on a given family of bioactive molecules

MAJOR ISSUES

1.       Lines 29-31. Please provide an appropriate reference.

2.       Paragraph 2: I agree with the authors in focusing on polyphenols as key bioactive molecules. Nevertheless, I suggest to modify the paragraph by explaining the rationale of their choice and provide a homogenous focus on the molecules that will be considered for the following sections (see 34836087). Accordingly, the table 1 should be re-edited.

3.       Paragpraph 3: I suggest to provide a conclusion to this paragraph summarizing the principal observations linking the respiratory diseases to inflammation, oxidative stress, microbial infections, etc (e.g., for COPD see 35326114). This could be preparatory for a better understanding of the valuable effects played by olive bioactive compounds

4.       Figure 3 and table 3: If the authors provide figures categorizing the in vivo and in vitro evidence, I suppose to find this approach also in text arguing the evidence addressed in paragraph 4 and 5.

5.       Paragraph 6: why did the authors make a focus on lung cancer? If so, it must be addressed more in detail,

6.       Lines 428-429: the authors stated “In terms of clinical trials, only two trials were conducted on olive effects on the respiratory system”. Are they sure of this? Please check at https://clinicaltrials.gov/.

MINOR ISSUES

1.       Please refer to Olive bioactive molecules in each paragraph title.

2.       Table 3 must be formatted properly. The references column should be moved at the end of the table. The authors should provide the information of “dosage” only in one column and always in this they will chose

Round 2

Reviewer 1 Report

That authors have addressed my concerns. I don't have any more comments. 

Author Response

We really appreciate your input and effort in improving our manuscript. Thank you so much. 

Reviewer 2 Report

The authors addressed the issues moved. However a re-editation of the tables must be done before review pubblication.

Author Response

Dear reviewer our humble apologies to you. We have formatted the tables according to the template. We have attached a PDF version as well in order to avoid distortion. Thank you.
